# Therapeutic Effect of Enzymatically Hydrolyzed Cervi Cornu Collagen NP-2007 and Potential for Application in Osteoarthritis Treatment

**DOI:** 10.3390/ijms241411667

**Published:** 2023-07-19

**Authors:** Ha-Rim Kim, Seung-Hyeon Lee, Eun-Mi Noh, Bongsuk Choi, Hyang-Yim Seo, Hansu Jang, Seon-Young Kim, Mi Hee Park

**Affiliations:** 1Jeonju AgroBio-Materials Institute, Wonjangdong-gil 111-27, Deokjin-gu, Jeonju-si 54810, Jeollabuk-do, Republic of Korea; poshrim@jami.re.kr (H.-R.K.); sh94@jami.re.kr (S.-H.L.); jingle1234@hanmail.net (E.-M.N.); 2Hanpoong Nature Pharm Co., Ltd., 91, Techno Valley 2-ro Bongdong-eup, Wanju-gun 55314, Jeollabuk-do, Republic of Korea; bongsuk333@naver.com; 3Jeonbuk Institute for Food-Bioindustry, Wonjangdong-gil 111-18, Deokjin-gu, Jeonju-si 54810, Jeollabuk-do, Republic of Korea; hiseo@jif.re.kr (H.-Y.S.); jhs@jif.re.kr (H.J.)

**Keywords:** enzyme hydrolyzation, cervi cornu collagen, NP-2007, osteoarthritis, monosodium iodoacetate

## Abstract

Cervi cornu extracts have been used in traditional medicine for the treatment of various disorders, including osteoporosis. However, since it is not easy to separate the active ingredients, limited research has been conducted on their functional properties. In this study, we extracted the low-molecular-weight (843 Da) collagen NP-2007 from cervi cornu by enzyme hydrolyzation to enhance absorption and evaluated the therapeutic effect in monosodium iodoacetate-induced rat osteoarthritis (OA) model. NP-2007 was orally administered at 50, 100, and 200 mg/kg for 21 days. We showed that the production of matrix metalloproteinase-2, -3, and -9, decreased after NP-2007 treatment. The levels of tumor necrosis factor (TNF)-α, interleukin (IL)-1β, IL-6, and prostaglandin E2 were also reduced after treatment of NP-2007. Furthermore, the administration of NP-2007 resulted in effective preservation of both the synovial membrane and knee cartilage and significantly decreased the transformation of fibrous tissue. We verified that the treatment of NP-2007 significantly reduced the production of nitric oxide and pro-inflammatory cytokines including TNF-α, IL-1β, and IL-6 in lipopolysaccharides-stimulated RAW 264.7 cells by regulation of the NF-kB and MAPK signaling pathways. This study indicates that NP-2007 can alleviate symptoms of osteoarthritis and can be applied as a novel treatment for OA treatment.

## 1. Introduction

Osteoarthritis (OA) is a prevalent degenerative joint disease that primarily affects cartilage and is accompanied by inflammation in the synovial tissue [1,2]. OA occurs when there is an imbalance between the synthesis and degradation of proteoglycans and collagen in cartilage cells [3]. Common manifestations of OA encompass gradual deterioration of articular cartilage, mild inflammation in the intra-articular and periarticular tissues, and occasionally the development of osteophytes and subchondral cysts in the affected joints [2], eventually leading to pain and disability, reducing the quality of life [4]. Therefore, the aims of OA treatment are to reduce pain, alleviate the disease, or prevent further cartilage damage [5]. The recognition of inflammation as a crucial factor in the development and progression of arthritis has led to the emergence of anti-inflammatory agents as a promising therapeutic strategy for the treatment of this condition. Several studies have highlighted the involvement of inflammation in the pathogenesis of arthritis, thereby emphasizing the importance of targeting inflammatory processes for effective management of the disease [6]. Although non-steroidal anti-inflammatory drugs (NSAIDs) are commonly prescribed for the treatment of OA, their long-term use can be associated with various side effects. These include an increased risk of gastrointestinal complications such as bleeding and stomach ulcers, particularly when used over an extended period of time [7]. Therefore, there is a need for alternative treatment approaches that provide effective pain relief and anti-inflammatory effects without the potential risks associated with long-term NSAID use, and recent studies have focused on the development of safer and more effective treatment strategies [8]. 

Cervi cornu is ossified antler and has various pharmacological effects. Deer antler has great potential in treating many diseases such as bone injuries, neurodegenerative diseases, tumors, and inflammatory conditions [9,10,11,12,13]. Zhang et al. showed that antler stem cells had an effect on wound healing and bone regeneration [9]. Liu et al. identified that velvet antler methanol extracts ameliorate Parkinson’s disease by inhibiting oxidative stress and neuroinflammation [10]. Moreover, it was reported that deer growing antlers had antitumor activity for the treatment of malignant gliomas [11]. Hu et al. showed that polypeptides from sika antler plate have antitumor activities in breast cancer cells [12], and Wu et al. showed that pilose antler polypeptides ameliorate hypoxic–ischemic encephalopathy by activated neurotrophic factors and the SDF1/CXCR4 axis in rats [13]. In addition, velvet antler, which refers to the cartilaginous antler in its precalcified stage, has a long history of use in traditional Chinese medicine and in functional foods. For thousands of years, it has been recognized for its potential in promoting overall wellness and maintaining good health [14]. Recent studies have highlighted the pharmacological potential of velvet antler [15], wound healing [9], and anti-cancer effects [16]. In particular, anti-inflammatory peptides consisting of 68 amino acid residues (7.2 kDa) have been identified from the antlers of *Cervus nippon* Temminck [17]. However, there is a lack of extensive information regarding the specific anti-inflammatory peptides derived from protein hydrolysates of velvet antler, including their primary structure. Recent research findings have indicated that the use of tortoise shells and antler gelatins can improve serum insulin-like growth factor levels and stimulate the proliferation of osteoblasts [18]. Although cervi cornu extracts have various functional benefits, the effect of the low-molecular-weight collagen derived from deer antlers has not been reported. Hydrolyzed collagens consist of amino acids and peptides, including dipeptides and tripeptides, that are resistant to intracellular hydrolysis and systemic hydrolytic enzymes, preventing their degradation by peptidases. As a result, these peptides have high bioavailability, enabling them to enter the bloodstream and accumulate in cartilage tissue. They stimulate chondrocytes and induce the synthesis of the cartilage extracellular matrix, promoting cartilage health and regeneration. The bioavailability of amino acids and peptides derived from hydrolyzed collagens plays a crucial role in understanding the effects of these products on joint health. Studies have shown that peptides resistant to intracellular hydrolysis have lower molecular weights, which facilitates their absorption in the intestines. This enhanced intestinal absorption leads to the higher bioavailability of these peptides, allowing them to exert their effects at the articular level and potentially promote joint health and function [19].

Here, we found that enzyme-hydrolyzed cervi cornu collagen NP-2007 alleviated monosodium iodoacetate (MIA)-induced OA. Therefore, we investigated the anti-inflammatory effects and molecular mechanisms of NP-2007 using an MIA-induced model and evaluated its potential for the treatment of OA. Moreover, we examined the anti-inflammatory properties of NP-2007 by evaluating its effects on RAW 264.7 cells that were stimulated with lipopolysaccharides (LPS). The findings of this study offer valuable insights into the treatment of OA, presenting potential strategies that could enhance the quality of life for individuals affected by this condition. By elucidating the underlying mechanisms and exploring potential therapeutic approaches, this research holds promise for advancing OA care and improving patient outcomes.

## 2. Results

### 2.1. NP-2007 Mitigated Articular Cartilage in MIA-induced OA Rats

Following an injection of MIA to induce OA, the SD rats were orally administered with NP-2007 (50, 100, 200 mg/kg) for a three-week period (Figure 1A). After sacrificing the rats, we applied microcomputed tomography (micro-CT) to examine the effects of NP-2007 on the architecture of knee joints and to implement a morphological evaluation of the articular cartilage. Compared with the control group, the MIA-treated group exhibited more deterioration of the distal femur bone architecture and decreased cartilage volume (Figure 1B). However, NP-2007 treatment efficiently preserved the architecture of the femur bone and enhanced cartilage volume.

Additionally, histopathological analysis was conducted on the knee cartilage using hematoxylin and eosin (H&E) and safranin O staining to assess the potential alleviating effect of NP-2007. The H&E staining demonstrated the presence of well-preserved cartilage in the control. However, the negative control (NC) group exhibited damaged cartilage and an increased infiltration of inflammatory cells. In contrast, the NP-2007-treated groups exhibited recovered cartilage and reduced inflammatory cell infiltration (Figure 1C). Moreover, safranin O staining revealed decreased proteoglycan volumes in the MIA-treated group, which was ameliorated by NP-2007 administration (Figure 1D).

These results indicated that NP-2007 exerts anti-inflammatory effects and protects the knee joint structure.

### 2.2. NP-2007 Alleviated Inflammation in MIA-Induced OA Rats

The serum levels of inflammatory factors and cytokines were evaluated in MIA-induced arthritic rats to investigate their effects on inflammation. The ELISA results demonstrated that the MIA-induced OA rats had significantly elevated serum levels of TNF-α, IL-1β, IL-6, and PGE2 and that NP-2007 administration effectively reversed these changes (Figure 2). These findings suggested that in rats with OA, NP-2007 might alleviate inflammation by suppressing inflammatory factors. 

### 2.3. NP-2007 Modulated the Matrix Metalloprotease Levels in MIA-Induced OA Rats

Inflammation in the joint is triggered by inflammatory factors and pro-inflammatory cytokines, which promote the excretion of cartilage-degrading enzymes such as matrix metalloproteinases (MMPs), leading cartilage degradation to accelerate [20]. So, we analyzed the serum levels of MMP-2, -3, and -9 to determine whether NP-2007 directly affects MMP expression. The serum levels of MMP-2, -3, and -9 significantly increased after MIA injection; however, NP-2007 effectively suppressed MMP-2 and -3 levels in MIA-induced OA rats (Figure 3). For the MMP-9 level, a significant difference was observed only at 200 mg/kg. These results suggested that NP-2007 can decrease MMP production and the degradation of cartilage and bone in arthritic joints.

### 2.4. NP-2007 Suppresses NF-κB Signaling in LPS-induced RAW 264.7 Macrophages

To identify the molecular mechanisms related to inflammation, we used RAW 264.7 cells and confirmed whether NP-2007 could mitigate LPS-induced inflammatory effects. The LPS-induced model in RAW 264.7 murine macrophages is a common model of inflammation used in in vitro anti-inflammatory studies [21] because the macrophage is an important immune cell and plays a pivotal role during inflammation in host defenses against pathogen infection. So, we used this model to study the anti-inflammatory effect of NP-2007. 

First, to evaluate cytotoxicity, RAW 264.7 cells were treated with various concentrations of NP-2007, and the MTS assay was implemented to evaluate cell viability. Cell viability was not significantly impacted by NP-2007 at concentrations up to 1000 μg/mL (Figure 4A). The NP-2007 concentrations of 100, 250, and 500 μg/mL were employed in the investigations that followed. 

To evaluate the inhibitory effects of NP-2007 on LPS-induced NO production, various doses of NP-2007 were applied to RAW 264.7 cells, and they were then incubated for 18 h with or without LPS (1 μg/mL). The production of NO was significantly reduced in a dose-dependent manner by NP-2007, as shown in Figure 4B. NP-2007 substantially inhibited LPS-induced protein expression of iNOS, one of the major enzymes responsible for NO production (Figure 4C,D). NP-2007 also attenuated LPS-induced COX-2 protein expression. To investigate the molecular mechanism via which NP-2007 inhibits inflammatory responses, we evaluated the protein expression of phosphorylated NF-κB, which is a critical transcription factor that regulates the production of pro-inflammatory cytokines. As shown in Figure 4C,D, LPS treatment stimulated the phosphorylation of NF-κB in RAW 264.7 cells. However, this effect significantly decreased in a dose-dependent manner with NP-2007. 

These findings suggested that NP-2007 treatment affects the inflammation reaction in RAW 264.7 cells’ inflammatory response by altering the NF-κB signaling pathway.

### 2.5. NP-2007 Suppresses Pro-Inflammatory Cytokine Production and MAPK Signaling in LPS-Induced RAW 264.7 Macrophages

To investigate whether NP-2007 could mitigate LPS-induced pro-inflammatory cytokine levels, RAW 264.7 cells were pretreated with NP-2007 (100, 250, or 500 μg/mL) for 1 h before being stimulated with LPS (1 μg/mL) for 18 h. In contrast to untreated cells, LPS-treated RAW 264.7 macrophages created pro-inflammatory cytokines (TNF-a, IL-1β, and IL-6), as demonstrated by ELISA analyzes (Figure 5A–C). However, NP-2007 application significantly and independently lowered the levels of cytokines IL-6 and IL-1β released by LPS-treated macrophages.

In comparison to the LPS-only control, NP-2007 significantly enhanced the amount of the anti-inflammatory cytokine IL-10 that is produced when LPS is present (Figure 5D). These findings suggested that NP-2007 improved IL-10 production and attenuated LPS-induced pro-inflammatory cytokine production in RAW 264.7 macrophages.

TNF-α, IL-1β, and IL-6 cytokines cause inflammation by activating critical proteins in signaling pathways like NF-kB and MAPK. We further investigated whether NP-2007 inhibited LPS-induced MAPK signaling. As MAPK is stimulated with phosphorylation, we performed Western blotting to evaluate the levels of phosphorylation of JNK, ERK, and p38 MAPK in RAW 264.7 macrophages using Western blotting. In cells treated with LPS, increased levels of phosphorylation of all MAPK signals were observed compared with the normal (N) group. NP-2007 markedly inhibited the LPS-induced phosphorylation of ERK, JNK, and p38 (Figure 6A,B).

These results suggested that NP-2007 could exert its protective effects against LPS-induced RAW 264.7 macrophages by modulating MAPK and NF-κB signaling. 

## 3. Discussion

In this study, we demonstrated, for the first time, that a low-molecular-weight collagen from deer horn, NP-2007, is effective against MIA-induced OA. Although traditionally considered a non-inflammatory joint disease due to its primary association with cartilage degeneration, it is now well established that inflammation plays a critical role in the etiology of OA. Inflammation contributes to various pathological changes observed in OA, highlighting the importance of understanding and targeting the inflammatory processes for effective management and treatment of the disease [22]. Hence, the identification of inflammation and pain as potential therapeutic targets opens up new avenues for the development of novel drugs aimed at addressing these specific aspects of OA. By targeting inflammation and pain mechanisms, researchers and clinicians can work towards the development of more effective and targeted treatment strategies to alleviate symptoms, slow down disease progression, and improve the quality of life for individuals with OA. 

The primary objectives of OA treatment include managing pain, minimizing cartilage damage, slowing down the progression of the disease, relieving symptoms, and improving or maintaining functional abilities. By addressing these goals, healthcare professionals aim to enhance the overall well-being and quality of life of individuals affected OA [23]. Despite the known limitations in the therapeutic efficacy and the potential for side effects such as cardiovascular risk, gastrointestinal upset, diarrhea, vomiting, nausea, and renal toxicity, common treatments for OA such as acetaminophen, NSAIDs, and opioids have remained largely unchanged in recent decades [23,24]. Hence, there is a critical need to explore and develop natural products as potential treatments for OA. In recent research, various traditional medicinal herbs have been extensively studied as potential sources for alternative and complementary approaches in managing OA. These studies aim to explore the use of dietary supplements, functional foods, and nutraceuticals derived from these herbs, with the goal of reducing pain and slowing down the progression of the disease [25,26]. In this study, we used a low-molecular-weight collagen, NP-2007, from deer horns and identified its anti-inflammatory properties. 

Different collagen preparations include insoluble undenatured native collagens, soluble native collagens, denatured collagens (gelatins), collagen hydrolysates, and collagen peptides. Each product is obtained through specific manufacturing processes, resulting in variations in structure, composition, and properties [27]. Collagen hydrolysates, formed through chemical or enzymatic hydrolysis, lack the triple-helix structure and consist of amino acids and peptides. Their composition varies depending on the collagen source and hydrolysis method used, such as pepsin, alcalase, and papain, with molecular weights typically ranging from 1 kDa to 10 kDa [28]. We used the cervi cornu-derived collagen hydrolysate with a molecular weight of 843 Da. Similar to other approved joint health functional materials, we studied the joint health in an MIA-induced OA model. MIA is a compound that is commonly used in experimental models to induce OA-like pathology in rodents. By inhibiting the enzyme glyceraldehyde-3-phosphatase, MIA causes chondrocyte apoptosis and leads to cartilage degeneration [29]. This MIA-induced OA model is valuable for studying the mechanisms underlying OA development and evaluating potential therapeutic interventions for the disease. This model triggers inflammatory pain and therefore is the most commonly used model, especially for pain treatment [30,31]. Moreover, when inflammation is activated, NF-κB activated in macrophages induces the release of interleukins, which causes caspase-mediated chondrocyte damage and arthritis [32]. Through MIA induction, the joint is degraded, and MMPs are upregulated. Several researchers have used this model and performed mechanistic studies to approve functional health materials. For example, AyuFlex^®^, an extract from *Terminalia chebula* Retzins, is effective in the treatment of MIA-induced OA through significant inhibition of OA symptoms such as oxidative stress, cartilage damage, and changes in cytokine and MMP regulation in arthrodial cartilage [33]. Additionally, *Andrographis paniculate* extracts were approved for the treatment of joint health in an MIA-induced OA model by decreasing the levels of inflammatory cytokines, namely, IL-1β, IL-6, and TNF-α, and the joint tissue degradation enzyme, MMP [34]. Similar to other approved joint health functional materials, we studied joint health in an MIA-induced OA model. We confirmed that the joint was degraded via MIA, and these effects were reversed upon NP-2007 treatment. Histopathological studies also revealed that the pathogenesis was reversed upon NP-2007 treatment.

Numerous research studies have consistently shown elevated levels of inflammatory cytokines, including IL-6, IL-1β, and TNF-α, in the knee joints of individuals with OA. The stimulation of cartilage tissue by pro-inflammatory cytokines has been associated with various structural changes that are characteristic of OA. These findings highlight the crucial role of inflammation in the pathogenesis and progression of OA, providing insights into potential therapeutic targets for managing the disease [35,36,37,38]. Additionally, individuals with OA often experience joint cartilage loss and disruption of bone structure, leading to the manifestation of joint pain [39]. So, the objective of this study was to demonstrate and examine these biomarkers in order to gain a better understanding of the role of NP-2007 in the development and progression of OA. NP-2007 demonstrated inhibitory effects on the inflammatory response by reducing the expression of inflammatory mediators, such as COX-2 and iNOS, along with cytokines including TNF-α, IL-1β, and IL-6. In addition, NP-2007 decreased joint cartilage loss and disruption of bone structure by downregulating MMPs. Through our mechanistic investigation, we observed that NP-2007 exerts its effects by inhibiting the NF-kB and MAPK signaling pathways. Our findings highlight the potential of NP-2007 as a therapeutic agent for OA by targeting key pathways involved in the regulation of inflammation. 

Collectively, our findings suggested that NP-2007 has anti-inflammatory effects and may be useful for the treatment of OA and other inflammation-related diseases.

## 4. Materials and Methods

### 4.1. Low-Molecular-Weight Cervi Cornu Collagen (NP-2007) Preparation

NP-2007 was prepared by enzyme hydrolyzation of cervi cornu and commercialized as a food ingredient by Hanpoong Nature Pharm. Briefly, cervi cornu was boiled 9 times with purified water at 95~99 °C for 3 h. Then, the extracts were filtered and concentrated. The first hydrolysis was performed with neutralase, and the second hydrolysis was performed with flavorzyme. After hydrolysis, the extracts were vacuum-dried and used as a raw food material.

### 4.2. Cell Culture

RAW 264.7 macrophages were acquired from the American Type Culture Collection (Manassas, VA, USA) and cultivated at 37 °C in a 5% CO_2_ environment in Dulbecco’s modified Eagle’s medium (DMEM) with penicillin–streptomycin sulfate (100 units/mL and 100 g/mL) and 10% fetal bovine serum (Thermo Scientific, Waltham, MA, USA).

### 4.3. Cell Viability Assay

Cell viability was determined using the MTS assay. In 96-well culture plates, cells (1 × 10^4^ cells/well) were seeded and treated for 24 h with NP-2007 dissolved in DMSO. Then, 10 μL of MTS solution (Promega, Madison, WI, USA) was added to the cells in each well, and the cells were incubated for a further 4 h. At 490 nm, absorbance was determined using a microplate reader (Multiskan Go, Thermo Scientific, Waltham, MA, USA). The values of the control were considered 100% viable. 

### 4.4. Nitric Oxide (NO) Assay

Cells (5 × 10^5^ cells/well) were placed in 6-well plates and cultured for 18 h with NP-2007 (100, 250, and 500 μg/mL) and lipopolysaccharides (LPS) (1 μg/mL) for 18 h as previously described [40]. The culture medium was analyzed using colorimetric assay kits (iNtRON Biotechnology, Sungnam, Republic of Korea) following the manufacturer’s requirements. The established standard curve for sodium nitrite (NaNO_2_) was used to calculate the nitrite concentration. 

### 4.5. Animals

Male Sprague Dawley rats (7 weeks old) were purchased from Damul Science (Daejeon, Republic of Korea). The typical housing settings for rats were 22 ± 2 °C and 55 ± 5% humidity (12 h light/dark cycles). The rats were randomly divided into six groups. Before the experiments began, the rats were given a week to get used to the conditions in the laboratory and were allowed free access to food and drink. The Jeonju AgroBio-Materials Institute’s Animal Care Committee evaluated and approved the experimental procedure for this study, which strictly complied with the committee’s guidelines (JAMI IACUC 2022001).

### 4.6. Induction of Monosodium Iodoacetate (MIA)-Induced Osteoarthritis (OA) in Rats

Rats with MIA-induced arthritis were generated as previously described [41]. First, the rats were anesthetized with avertin (250 mg/kg intraperitoneally). Using an insulin syringe, MIA solution consisting of 3 mg of MIA dissolved in 50 μL of 0.9% saline was injected precisely into the intra-articular space of the right knees of rats using an insulin syringe. After MIA injection, the rats with MIA-induced OA were treated with NP-2007 at various doses (50, 100, and 200 mg/kg) of NP-2007 dissolved in water for 21 d. The positive control group was treated with 300 mg/kg methyl sulfonyl methane (MSM). Distilled water was orally administered to the N and NC groups.

### 4.7. Cytokine Assay

Cell supernatants and rat serum were prepared to measure cytokine concentrations. The enzyme-linked immunosorbent assay (ELISA) kits provided by R&D Systems were used for assessing the levels of matrix metalloproteinase (MMP)-9, interleukin (IL)-1β, IL-6, prostaglandin E2 (PGE2), and tumor necrosis factor (TNF)-α. The MMP-2 and MMP-3 concentrations were measured using Quantikine ELISA kits (Abcam, Cambridge, UK). Instructions provided by the manufacturer were followed for all measurements.

### 4.8. Immunoblotting

SDS-PAGE (8% or 10%) gels with polyvinylidene difluoride membranes (GE Healthcare, Little Chalfont, Buckinghamshire, UK) were used to resolve proteins (20 μg per lane). Primary antibodies (anti-iNOS (Cat#13120, 1:1000), anti-phospho-NF-κB (Cat#8242, 1:2500), anti-NF-κB (Cat#3033, 1:2500), anti-phospho-SAPK/JNK (Cat#4668, 1:2500), anti-SAPK/JNK (Cat#9252, 1:2500), anti-phospho-p44/p22 MAPK (Cat#9101, 1:2500), anti-p44/p22 MAPK (Cat#4695, 1:2500), anti-phospho-p38 MAPK (Cat#4511, 1:2500), anti-p38 MAPK (Cat#8690, 1:2500), and anti-β-Actin (Cat#3700, 1:10,000) antibodies from Cell Signaling Technology (Danvers, MA, USA) and anti-COX-2 (Cat#15191, 1:5000) antibody from Abcam (Cambridge, UK)) were incubated with the blots overnight at 4 °C. After blocking buffer washes, secondary antibodies combined with horseradish peroxidase were incubated with the blots for 1 h at room temperature. An enhanced chemiluminescence system (Bio-Rad, Munich, Germany) was used to detect antibody binding. Densitometric scanning (Amersham Imager 600; GE Healthcare) was utilized to evaluate all immunoreactive signals.

### 4.9. Histological Analysis

The microarchitecture of the knee joints was measured by a microcomputed tomography (micro-CT) system (Sky-Scan 1076; SkyScan, Aartselaar, Belgium). The tissues of the knee joints were fixed for 24 h in 4% paraformaldehyde. Sections were embedded in paraffin after fixation. 

The serial sections, measuring 4 μm in thickness, were mounted onto slides coated with silane and subsequently stained for the detection of proteoglycans with safranin O or for general histopathology with hematoxylin and eosin (H&E). A microscope using an optical system was employed to examine and photograph images of all stained specimens (Olympus, Tokyo, Japan).

### 4.10. Statistical Analyses

The data represent the means ± standard deviation (SD) derived from a minimum of three separate experiments. Student’s *t*-test was applied to compare parameters between two groups, and analysis of variance (ANOVA) followed by Duncan’s post hoc test were employed to compare parameters among the three groups. Statistics were judged significant at *p* < 0.05. 

## 5. Conclusions

We present novel findings indicating that NP-2007 exhibits inhibitory effects on MIA-induced OA. This is the first study to demonstrate the anti-inflammatory properties of NP-2007, which was shown to modulate the MAPK and NF-κB signaling pathways. These results provide valuable insights into the potential therapeutic applications of low-molecular-weight collagen in mitigating inflammation-associated conditions. Building upon the promising outcomes of our preclinical study, we are currently in the process of designing a clinical study to investigate the effects of low-molecular-weight collagen on patients diagnosed with OA. 

## Figures and Tables

**Figure 1 ijms-24-11667-f001:**
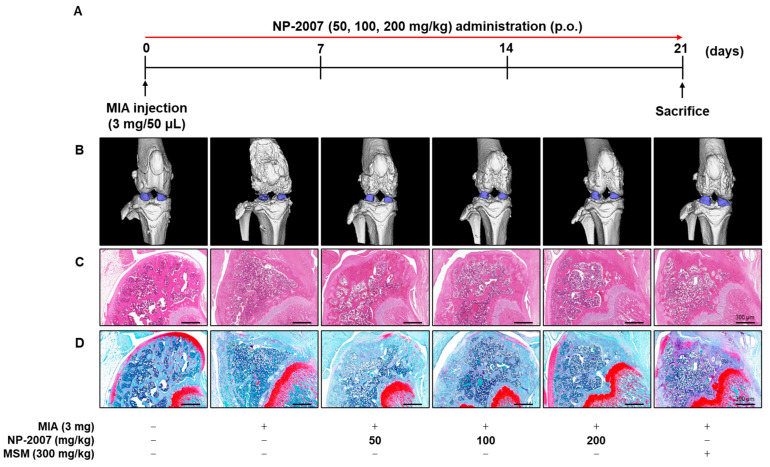
Effect of low-molecular-weight cervi cornu collagen (NP-2007) on morphological and histopathological changes from knee joint tissue of monosodium iodoacetate (MIA)-induced osteoarthritis (OA) rats. (**A**) Experimental scheme of MIA-induced OA model in rats. Vehicle, NP-2007 (50, 100, 200 mg/kg), or methyl sulfonyl methane (MSM) (300 mg/kg) were orally administered for three weeks. Morphological and histopathological analysis showing the effect of NP-2007 on knee joint tissue of MIA-induced OA rats. Representative (**B**) morphological image of the knee joint tissues measured using microcomputed tomography (micro-CT). The representative histopathological images of (**C**) hematoxylin and eosin (H&E) and (**D**) safranin O staining. Magnification: 40×.

**Figure 2 ijms-24-11667-f002:**
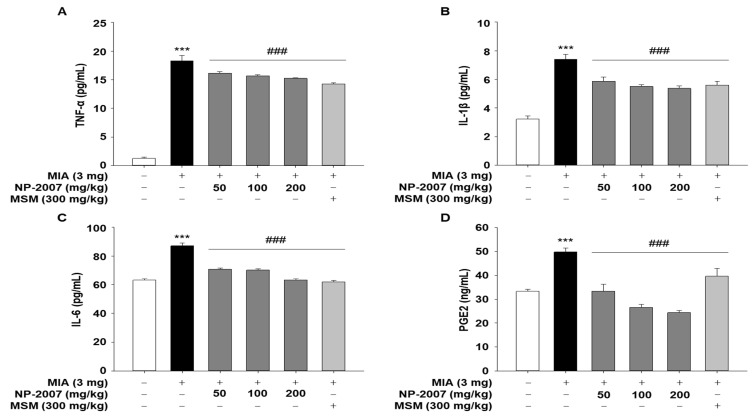
Effect of NP-2007 on inflammatory mediator and pro-inflammatory cytokine levels in serum of MIA-induced OA rats. ELISA was used to quantify the amount of (**A**) tumor necrosis factor (TNF)-α, (**B**) interleukin (IL)-1β, (**C**) IL-6, and (**D**) prostaglandin E2 (PGE2) in serum. All values represent the mean ± SD. Data were examined using Duncan’s multiple comparison test. *** *p* < 0.001 vs. control group; ### *p* < 0.001 vs. MIA-injected group.

**Figure 3 ijms-24-11667-f003:**
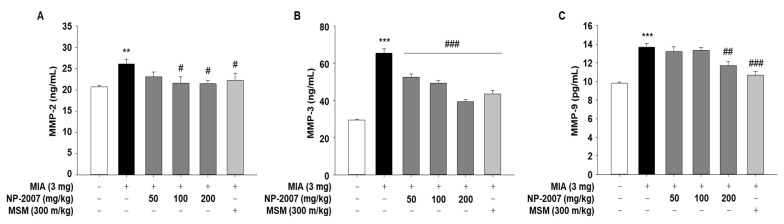
Effect of NP-2007 on matrix metalloproteinase (MMP) production in serum of MIA-induced OA rats. (**A**) MMP-2, (**B**) MMP-3, and (**C**) MMP-9 production were measured using enzyme-linked immunosorbent assays (ELISAs). All values represent the mean ± SD (*n* = 7). Data were examined using Duncan’s multiple comparison test. ** *p* < 0.01 and *** *p* < 0.001 vs. control group; # *p* < 0.05, ## *p* < 0.01, and ### *p* < 0.001 vs. MIA-injected group.

**Figure 4 ijms-24-11667-f004:**
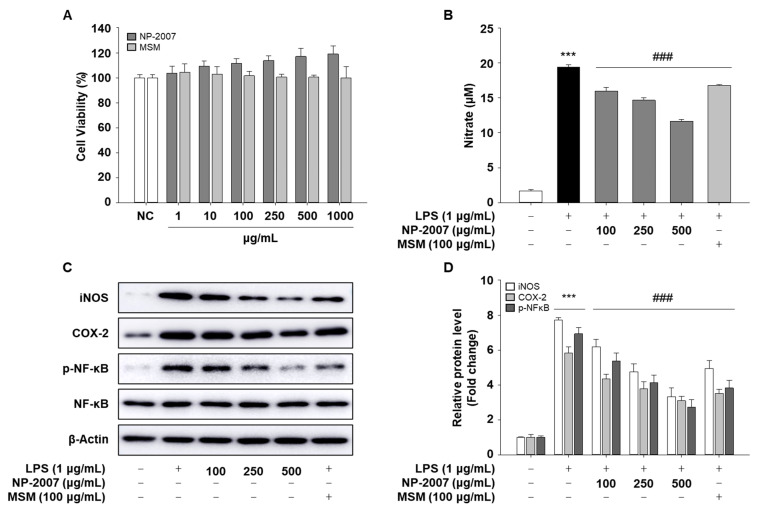
Inhibitory effect of NP-2007 on LPS-induced nitric oxide (NO) production and the NF-kB pathway in RAW 264.7 cells. (**A**) Effects of NP-2007 on RAW 264.7 cell viability. (**B**) NO production was determined in supernatants from RAW 264.7 cells. (**C**) Protein expression of iNOS, COX-2, and NF-kB signaling pathways were determined via Western blotting. SDS-PAGE effectively processed equivalent amounts of total proteins. (**D**) Quantitative analysis of blots. The densitometry results are represented as the relative protein band densities normalized to the β-actin level. All values represent the mean ± SD. Data were examined using Duncan’s multiple comparison test. *** *p* < 0.001 vs. control group; ### *p* < 0.001 vs. LPS group.

**Figure 5 ijms-24-11667-f005:**
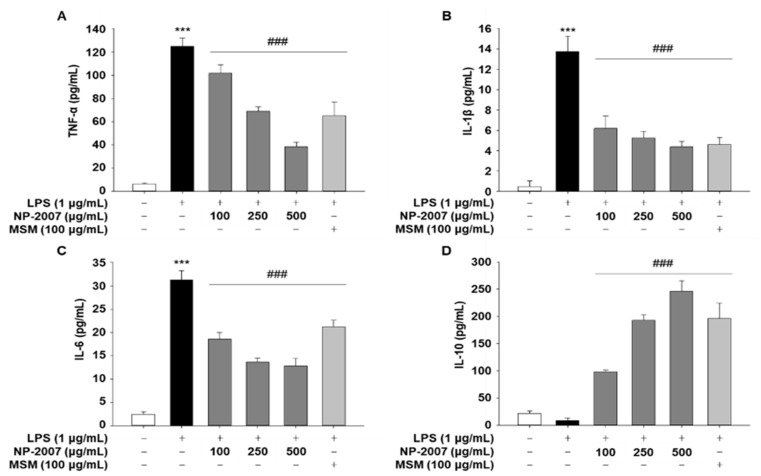
Inhibitory effect of NP-2007 on LPS-induced inflammatory cytokine production in RAW 264.7 cells. Concentrations of (**A**) tumor necrosis factor (TNF)-α, (**B**) interleukin (IL)-1β, (**C**) IL-6, and (**D**) IL-10 in RAW 264.7 macrophages. All values represent the mean ± SD. Data were examined using Duncan’s multiple comparison test. *** *p* < 0.001 vs. control group; ### *p* < 0.001 vs. LPS group.

**Figure 6 ijms-24-11667-f006:**
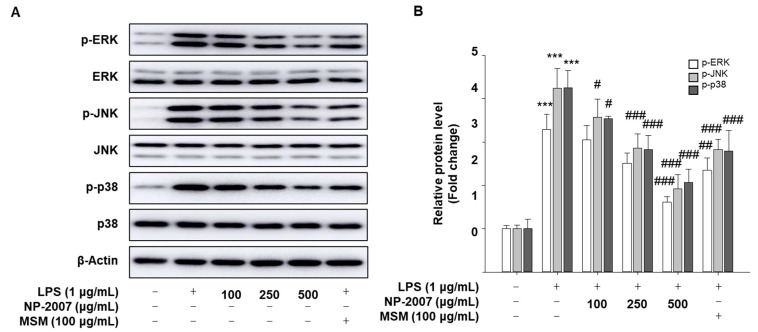
Inhibitory effect of NP-2007 on MAPK pathway in RAW 264.7 cells. (**A**) MAPK signaling pathways were determined via Western blotting. SDS-PAGE effectively processed equivalent amounts of total proteins. (**B**) Quantitative analysis of blots. The densitometry results are represented as the relative protein band densities normalized to the β-actin level. The total ERK, JNK, or p38 is shown as a loading control. All values represent the mean ± SD. Data were examined using Duncan’s multiple comparison test. *** *p* < 0.001 vs. control group; # *p* < 0.05, ## *p* < 0.01, and ### *p* < 0.001 vs. LPS group.

## Data Availability

The data presented in this study are available upon request.

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
