# Peer review of "Therapeutic Effect of Enzymatically Hydrolyzed Cervi Cornu Collagen NP-2007 and Potential for Application in Osteoarthritis Treatment"

_ijms, 2023, doi:10.3390/ijms241411667_

Round 1

Reviewer 1 Report

In the article: “Therapeutic effect of enzymatic hydrolyzed cervi cornu collagen NP-2007 and application in osteoarthritis”, the authors extracted the low molecular weight (843Da) collagen NP-2007 from cervi cornu by enzyme hydrolyzation, and evaluated the therapeutic effect in a monosodium iodoacetate (MIA)-induced rat osteoarthritis model.

Overall, this manuscript results very interesting, the authors clearly explain the rational of the study and discussed the topic point by point.

However, we would like to invite the authors  to clarify some minor points:

 1.       Please check the check punctuation and spaces;

2.       Concerning the ref 6 and 7, maybe should be opportune replace or add reference more recent;

3.       Page 2, lines 56-57; the authors said “Cervi cornu (Deer antler) has various pharmacological, analgesic and anti-inflammatory effects [9]. It has been reported that deer antlers prevent osteoporosis, and treat breast enlargement and mastitis [10, 11]”. Please try to better describe what it is because it is not enough clear;

4.       Materials and Methods; Immunoblotting; please add the primary antibodies used (brand, code) and the relative dilution used for the assay;

5.       Figure 2; except for PEG2, no so relevant differences between the treatments seem to be, why? Try to explain;

6.       Why the use of RAW264.7 cells is not reported in materials and methods section? Please insert in relative section;

7.       Why did you insult the cells with LPS  at 1 μg/mL? Please add the relative reference;

8.       Why did you observe the protein expression of biomarkers related to NO production and only NF-kB for inflammation? Are available other data? Maybe about biomarkers related to cartilage integrity; COMP-2 or Aggrecan?

9.       Figure 5; it is difficult to read and understand; maybe you should divide;

10.   Page 7, lines 269-273; the authors said “Numerous research studies have consistently shown elevated levels of inflammatory cytokines, including IL-6, IL-1β, and TNF-α, in the knee joints of individuals with OA. The stimulation of cartilage tissue by pro-inflammatory cytokines has been associated with various structural changes that are characteristic of OA. These findings highlight the crucial role of inflammation in the pathogenesis and progression of OA providing insights into potential therapeutic targets for managing the disease [31-33]”. Also the following reference should be useful; Vassallo V, Stellavato A, Russo R, Cimini D, Valletta M, Alfano A, Pedone PV, Chambery A, Schiraldi C. Molecular Fingerprint of Human Pathological Synoviocytes in Response to Extractive Sulfated and Biofermentative Unsulfated Chondroitins. Int J Mol Sci. 2022 Dec 14;23(24):15865. doi: 10.3390/ijms232415865. PMID: 36555507; PMCID: PMC9784855.

minor mistakes of spelling are here present 

Reviewer 2 Report

The manuscript submitted by Kim and colleagues demonstrates beneficial therapeutic effect of enzymatically hydrolyzed cervi cornu collagen on MIA-induced OA, as well as preventive and anti-inflammatory potential based on experiments with RAW264.7 cells. The manuscript is clear and well organized.

However, the following points should be considered before publication:

1. The title needs to be modified since the authors do not report particular apllication of NP2007 for treatment of OA patients. My suggestion for title is: "Therapeutic effect of enzymatically hydrolyzed cervi cornu collagen NP-2007 and potential for application in osteoarthritis treatment". 

 2. Improve the abstract. It should contain information about experiments with RAW264.7 cells. L16-17: the authors should clarify the sentence containing the following part: "absorption is not performed well". The study does not directly show preventive effect of NP-2007 on MIA-induced OA and thus, the last sentence of the abstract should be modified. For the same reason, the term "prevention" should be removed from L90 (end of Introduction) or the sentence should be modified.

3. Revision of terminology used in the manuscript:

- "enzymatic hydrolysed" should be corrected to enzymatically hydrolyzed;

- the term "cervi cornu" is used many times in the text (L15, L56, L69/70 etc.) but the authos should clarify that they mean cervi cornu extracts/different cervi cornu preparations or isolated from cervi cornu active compounds like polypeptides;

- L28: change "... and have applications..." with ... could have applications as a novel treatment for OA;

- L93: specify osteoarthritis instead of "arthritis";

- L116: revise the sentence; the use of "analyze" is improper;

- L247: change "collagen" with different collagen preparations or other more appropriate term;

- L292-293: revise "cervi cornu was extracted";

- L305: "cells were then given" can be changed to MTS solution was added to the cells/cell culture medium;

- L369: revise the sentence. The term low molecular weight collagen is more appropriate.

4. Define abbreviations first time they appear in the text (NC group, N group, micro-CT).

5. The Discussion section should be improved. More information/comparison with similar studies is needed.

These type of comments are included in the general comments and suggestions.

+ L256: correct "... an MIA-..."
